# Pinning Potential of the Self-Assembled Artificial Pinning Centers in Nanostructured YBa_2_Cu_3_O_7−x_ Superconducting Films

**DOI:** 10.3390/nano12101713

**Published:** 2022-05-17

**Authors:** Ion Ivan, Alina M. Ionescu, Daniel N. Crisan, Andreea Andrei, Armando Galluzzi, Massimiliano Polichetti, Jesus Mosqueira, Adrian Crisan

**Affiliations:** 1National Institute of Materials Physics, 405A Atomistilor Str., 077125 Magurele, Romania; ion.ivan@infim.ro (I.I.); alina.ionescu@infim.ro (A.M.I.); daniel.crisan@infim.ro (D.N.C.); 2National Institute for Laser, Plasma & Radiation Physics, 409 Atomistilor Str., 077125 Magurele, Romania; andreea.andrei@inflpr.ro; 3Department of Physics “E.R. Caianiello”, University of Salerno, Via Giovanni Paolo II, 132, 84084 Fisciano, Italy; agalluzzi@unisa.it (A.G.); mpolichetti@unisa.it (M.P.); 4QMatterPhotonics, Departamento de Física de Particulas, and Instituto de Materiais (iMATUS), Universidade de Santiago de Compostela, 15706 Santiago de Compostela, Spain; j.mosqueira@usc.es

**Keywords:** superconducting films, multilayers, nanoscale defects, artificial pinning centers, critical current density, pinning potential

## Abstract

For high-field power applications of high-temperature superconductors, it became obvious in recent years that nano-engineered artificial pinning centers are needed for increasing the critical current and pinning potential. As opposed to the artificial pinning centers obtained by irradiation with various particles, which is a quite expensive approach, we have studied superconducting samples having self-assembled defects, created during the sample fabrication, that act as effective pinning centers. We introduced a simple, straight-forward method of estimating the frequency-dependent critical current density by using frequency-dependent AC susceptibility measurements, in fixed temperatures and DC magnetic fields, from the positions of the maxima in the dependence of the out-of-phase susceptibility on the amplitude of AC excitation magnetic field. The results are compatible with a model that stipulates a logarithmic dependence of the pinning potential on the probing current. A mathematical derivation allowed us to estimate from the experimental data the pinning potentials in various samples, and in various DC magnetic fields. The resulted values indicate large pinning potentials, leading to very small probability of magnetic flux escaping the pinning wells, hence, leading to very high critical currents in high magnetic fields.

## 1. Introduction

Power applications of YBa_2_Cu_3_O_7−x_ (YBCO) films fabricated as long-length coated superconductors require low-cost methods for the preparation of superconducting films with high critical current density (*J*_c_). For high-magnetic fields applications (either to perform in high fields, or to generate high fields), high-*J*_c_ values require high density of defects for the prevention of the motion of magnetic flux induced by Lorentz force and thermal energy. Very soon after the discovery of YBCO, single crystals and epitaxial thin films have been fabricated and used for studies of fundamental properties of the material. It came as a great surprise that *J*_c_ in thin films was two orders of magnitude larger than in the single crystal, due to the very different nature and densities of nano-scale defects in the two morphologies. While in the single crystals there are low densities of weak, point-like defects due to oxygen vacancies, and few twinnings, in the YBCO films, there are a large number of different defects, occurring naturally during fabrication process, and they can be point defects, planar defects (e.g., stacking faults), misfit dislocations, out-of-plane misoriented grains, voids, anti-phase domain boundaries, surface roughness of *a*-axis grains, threading dislocations, in-plane misoriented grains and grain boundaries, twin boundaries, and precipitates [1].

However, especially in high magnetic fields, natural pinning centers are not enough, and/or not strong enough to overcome the very large Lorentz force, so addition of artificial pinning centers (nano-scale defects) is needed, through various approaches of self-assembled nano-engineering. First, methods like substrate decoration [2,3,4] or deposition from targets with small amount of a secondary phase nanoinclusions, e.g., with BaZrO_3_ (BZO) [5] or with Gd_2_Ba_4_CuWO_y_ [6], were creating one specific type of pinning centers (nanoparticles, nanorods, splayed linear defects composed of the secondary phase, or combination of the above). The improvement of *J*_c_ could be further increased by variations in the film architecture, e.g., using a multilayer structure in which YBCO layers are separated by nanoscale-thin insulating layer such as CeO_2_ [7] or by targeting several types of pinning centers by using double-doping, for example, with BZO and Y_2_O_3_ [8,9], or GdTaO_7_ and YBa_2_NbO_6_ [10]. More recently, large pinning force was reported by addition of mixed double perovskite Ba_2_Y(Nb/Ta)O_6_ in a GdBa_2_Cu_3_O_7−x_ superconducting matrix which resulted in an unusual pinning landscape consisting in Ba_2_(Y/Gd)(Nb/Ta)O_6_ segmented nano-rods parallel to the *c*-axis and (Y/Gd)_2_O_3_ plate-like nanoparticles parallel to the *a-b* plane [11]. Similar synergetic nano-scale pinning centers were obtained by the combination of substrate decoration with Ag nano-islands (that proved to promote columnar growth of YBCO [12]) with BZO-doping [13] and introduction of a nanoscale-thin layer of SrTiO_3_ in the middle of the BZO-doped YBCO film [14]. Electron microscopy showed BZO nano-rods and nanoparticles that act as effective pinning centers for magnetic flux lines in both directions: along the *c*-axis and in the *ab*-plane, as well as additional defects mostly in the *ab*-plane created by the STO nanoscale layer near the STO-YBCO interfaces, similar to the results in Ref. [15]. A comprehensive review on various types of pinning centers in various superconductors, using various approaches and various architectures can be found in Ref. [16]. More recently, an excellent review was devoted to processing and applications of high-temperature superconducting coated conductors, with emphasis on how the nanostructure of the HTS material impact wire performance across different application regimes [17]. However, most of the work in the field deals with improvement of critical current density and its anisotropy in respect with the angle between the sample and the applied field, determined mainly from transport measurements and/or magnetic hysteresis loops, and with the improvement on bulk pinning force extracted from the same magnetization loops.

In this work, we aim to investigate the pinning potential of the self-assembled artificial pinning centers in YBCO superconducting films with various architectures of nano-engineered defects, using a different approach, namely, frequency-dependent AC susceptibility response. Our study is performed at temperature of practical interest and in various applied DC fields, and to determine the pinning models that better describe our experimental data.

## 2. Materials and Methods

All six samples investigated in this work are similar, grown in completely the same conditions, as samples that were already characterized by DC magnetization loops, magnetic relaxation measurements, and angle-dependent magneto-transport measurements; their preparation and materials used is described extensively as follows: samples with artificial pinning centers induced by BZO and Ag in Refs. [12,13], sample with artificial pinning centers induced by BZO and Y_2_O_3_ in Ref. [8], and, respectively, sample with artificial pinning centers induced by BZO, Ag, and STO in Ref. [14]. In brief, the nano-engineered superconducting films were grown by Pulsed Laser Deposition (PLD) on SrTiO_3_ (STO) single-crystal substrates using a KrF excimer laser with 248 nm wavelength with pulse duration of 30 ns, repetition rate of 3–8 Hz, laser energy density of 0.7–1.7 J/cm^2^, laser energy between 240 and 266 mJ, and target–substrate distance of 5–6 cm, in a PLD chamber with computer-controlled multi-target carousel allowing sequential depositions from several targets. Pictures of the PLD equipment used for the growth of the majority of samples (all except for the one containing Y_2_O_3_) can be found in Ref. [18]. Substrate temperatures were either 780 °C or 800 °C and the oxygen partial pressure during deposition was 450 mTorr. After the deposition, the films were cooled down in partial oxygen atmosphere of 450 Torr at the rate of 8 °C/min. For YBCO films with Ag substrate decoration, the same conditions for YBCO growth have been used. Prior to YBCO deposition, Ag nanodots were grown, also by PLD, at a substrate temperature of 400 °C in vacuum. For Ag/YBCO quasi-superlattices, the same conditions for YBCO and Ag were used, by alternating YBCO layers and Ag quasi-layers several times. Best synergetic pinning centers were obtained by using a YBCO target with 4% BZO nanoinclusions, the higher temperature of 800 °C combined with a lower repetition rate of 3 Hz led to self-assembling of BZO into nano-rods, while 780 °C substrate temperature and repetition rate of 8 Hz led to both BZO nanoparticles and nanorods. Other synergetic pinning centers were obtained by adding Y_2_O_3_ nanoparticles, or a thin layer of STO sandwiched between BZO-doped YBCO layers. Additional information about the measured samples, most significant being the thickness, will be given in the Results section.

Frequency-dependent critical current density was estimated from the out-of-phase (imaginary) AC susceptibility χ″ response as function of the AC field amplitude *h*_ac_, at fixed temperatures *T* (77 K and 65 K) and fixed applied DC magnetic fields *H*_DC_ (between 3 and 14 T, in steps of 1 T), measured on a *Quantum Design* Physical Properties Measurement System PPMS in μ_0_*H*_DC_ up to 14 T, AC field amplitude *h*_ac_ up to 16 Oe, and AC field frequency *f* between 47 and 9997 Hz. At a given temperature, for several *H*_DC_, χ″(*h*_ac_) dependence may show a peak in the experimental window of our measurements, its position *h** representing the AC field of full penetration of the perturbation in the center of the sample, which can be correlated with the critical current density of the specimen [19]. By considering that the sample has one dimension much smaller than the other two and that it is characterized in perpendicular field configuration (magnetic field perpendicular to the film surface), from the Brandt approach on the flux penetration [20], the AC field value of χ″(*h*_ac_) peak, *h**, can be related to the critical current density by the equation
(1)$$J_{c}=\frac{h^{*}}{\alpha d}$$
where *d* is the film thickness and *α* is a constant between 0.8 and 0.9 that depends slightly on the geometry (square, circular disc, rectangle). In our case, the samples are squares with dimension 5 mm each side, and the thickness (which enters in Equation (1)) is indicated in the case of each sample. It is well established that, with the decrease of the AC field frequency *f*, the time available for a magnetic flux line (vortex) to escape the potential well of the pinning center increases, so the probability of escaping increases; hence, the critical current density will decrease with decreasing frequency. The frequency dependence of the peak position *h**(*f*) can provide information about the strength of the pinning center, the pinning potential *U*_0_. There are few models of pinning that relate the critical current density to the time-scale (frequency) and the potential well of the pinning centers. The Anderson–Kim model [21] assumes that flux creep occurs due to thermally-activated jumps of isolated bundles of flux lines between two adjacent pinning centers, the jump being correlated for a bundle of vortices of volume *V*_c_ (correlated volume) due to the interaction between vortices. In the absence of transport current or excitation current induced by an alternating magnetic field (i.e., zero Lorentz force), the bundle is placed in a rectangular potential well of height *U*_0_. Due to the thermal energy, there are jumps over the barrier with a frequency *f* = *f*_0_ × exp(−*U*_0_/*k*_B_T). When the Lorentz force due to a current density *J* is present, the potential well becomes asymmetric, with a decreased height in the direction of Lorentz force for forward jumps *U*_f_ = *U*_0_ (1 − *J*/*J*_0_) and an increased height in the opposite direction for backwards jumps *U*_b_ = *U*_0_ (1 + *J*/*J*_0_), where *J*_0_ is the critical current in the absence of the Lorentz force. It can be seen that the dependence of the pinning potential on the probing current is a linear one. For currents high enough (necessary for practical applications), the probability of backward jumps is much smaller than the probability of forward jumps, and critical current density is related to the frequency and to the pinning potential by the equation
(2)$$J_{c}=J_{0}\left[1+\frac{k_{B}T}{U_{0}}ln\left(\frac{f}{f_{0}}\right)\right]$$
where *f*_0_ is an attempt macroscopic frequency of about 10^6^ Hz.

Another well-established model is the Larkin–Ovchinnikov collective weak pinning model [22], which considers the cooperative aspects of vortex dynamics in which the formation of the vortex lattice will be a result of a competition between the vortex–vortex interaction which tends to place a vortex on a lattice point of a periodic hexagonal/triangular lattice (Abrikosov lattice) and the vortex–pinning center interaction which tends to place a vortex on the local minimum of the pinning potential. Vortex–vortex interaction promotes global translational invariant order, while vortex–pinning center interaction tends to suppress such long-range order if pinning potential varies randomly. In this model, the pinning potential has a power dependence on the probing current *U*(*J*) = *U*_0_(*J*_c_/*J*)^μ^, with μ ≤ 1, leading to a frequency dependence of the critical current density of the form
(3)$$J\left(f\right)=J_{c}{\left(\frac{k_{B}T}{U_{0}}ln\frac{f_{0}}{f}\right)}^{\frac{-1}{\mu }}$$

A logarithmic dependence of the pinning potential on probing current was proposed by Zeldov and coworkers [23], *U*_eff_ = *U*_0_*ln*(*J**/*J*), based on magneto-resistivity and current-voltage characteristics of YBCO films. Such dependence was shown to be consistent with a potential well having a cone-like structure exhibiting a cusp at its minimum and a broad logarithmic decay with the distance [24].

## 3. Results

The samples described in the previous section were investigated by measuring the dependence of the out-of-phase susceptibility response χ″ as function of the amplitude of the excitation AC field *h*_AC_, at various frequencies *f* of the excitation AC field, in fixed temperatures of interest (77 K, 65 K), and in various applied DC magnetic fields. In the cases when χ″(*h*_AC_) have a maximum at *h**, Equation (1) allowed us to determine the frequency-dependent *J*_c_. Numerical analysis of the data in the frame of the above-mentioned models allowed to find the right model for the pinning potential, and to estimate its value in each case.

### 3.1. Frequency-Dependent Out-of-Phase Susceptibility Response

An example of such χ″(*h*_AC_) measurement is shown in Figure 1, for the bi-layer film composed of two BZO-doped YBCO films of 1.5 μm thickness, grown on substrates decorated with Ag nano-islands, separated by a nanolayer (15 nm) of insulating STO, at 77 K, in a DC field of 7 T, and at various frequencies between 85 and 9997 Hz, the steps between the measurement frequencies being equidistant in a logarithmic scale.

It can be clearly seen that the position of the maximum *h** shifts towards higher values with increasing frequency. This is related to the fact that, at higher frequency, the timescale for the vortex jump out of the potential well is smaller; hence, the probability of escape is smaller, leading to a higher critical current density with increasing frequency. Such measurements were performed on all samples investigated in this study, mainly at the liquid nitrogen temperature of 77 K, and, in few cases, also at 65 K, a temperature of interest for nitrogen gas closed-cycle cryo-coolers. It is important to mention that, for a given film, at a given temperature, only a limited range of applied DC magnetic fields allow us to obtain the maxima in χ″(*h*_AC_). As an example, for a 1 μm-thick films, a maximum position at 10 Oe (maximum available *h*_AC_ being 16 Oe in the *Quantum Design* PPMS) represents a critical current density of approximately 10^5^ A/cm^2^, as resulted from Equation (1). So, for this thickness, critical current densities larger than that cannot be determined using this method. A closer look at Equation (1) reveals that for thinner films, the field range may be larger, while for thicker films, only few large DC fields allow the estimation of the critical current density with this experimental limitation on the maximum *h*_AC_.

### 3.2. Frequency-Dependent Critical Current Density: Results and Models

Measurements similar to those in Figure 1 were performed for all the analyzed samples, in several applied DC fields. In some cases maxima in χ″(*h*_ac_) could be observed, and, using Equation (1), we have estimated the frequency-dependent critical current density (with coefficient α = 0.9). Figure 2 shows the dependence of the critical current density *J*_c_ of the 0.4 μm-thick film of YBCO doped with 4% wt. BZO and 2% at. Y_2_O_3_, as function of AC field frequency *f*, at 77 K and in a DC field of 6 T, in a double-logarithmic scale, where *f*_0_ in the reformulation of the *x*-axis is a macroscopic attempt frequency of flux jumps outside the pinning centers of about 10^6^ Hz [25] (full symbols).

It can be clearly seen that the attempted fits with both “classical” models in the literature, Anderson–Kim model (Equation (2)) or collective pinning model (Equation(3)), are not successful. Experimental points lay quite on a straight line in the double-logarithmic plot. Similar results were obtained for all samples, Figure 3 (a and b being other two examples (one thinner film, and one thicker, over 1 μm).

For all the investigated samples, the experimental *J*_c_, in logarithmic scale, decreases linearly with *ln*(*f*_0_/*f*). In the following sub-chapter, we will show that such dependence is compatible with the logarithmic dependence of the pinning potential on the probing current, proposed by Zeldov et al. [22].

### 3.3. Pinning Potential from Frequency-Dependent Critical Current Density

We have seen that our experimental data are very well described by
(4)$$lnJ_{c}=a-b\left[ln\left(\frac{f_{0}}{f}\right)\right]$$
which can be rewritten as *ln**J* = *ln**J*_0_ − *bln*(*t*/*t*_0_), where *t* = 1/*f*, *t*_0_ = 1/*f*_0_ and *J*_0_ is the critical current density in the absence of thermally-activated flux creep (*T* = 0 K) or, by using the properties of logarithm function:(5)$$J=J_{0}{\left(\frac{t}{t_{0}}\right)}^{-b}$$

In an inductive circuit, the voltage *V* is proportional to the time derivative of the current, and, from Equation (5), is given by
(6)$$V\propto \frac{dJ}{dt}=-bJ_{0}t_{0}^{-1}{\left(\frac{t}{t_{0}}\right)}^{-\left(b+1\right)}$$

At the same time, the voltage (dissipation) is proportional to the probability of the flux jump/creep over the pinning barrier, which is an Arrhenius-type thermally-activated process, and, considering the logarithmic dependence of the effective pinning potential proposed by Zeldov,
(7)$$U_{eff}=U_{0}ln\left(\frac{J^{*}}{J}\right)$$
the voltage dependence on the probing current will be
(8)$$V\propto e^{-\frac{U_{eff}}{k_{B}T}}=e^{\left(\frac{-U_{0}}{k_{B}T}\right)ln\left(\frac{J^{*}}{J}\right)}$$

Comparing Equations (6) and (8), one has:(9)$$-bJ_{0}t_{0}^{-1}{\left(\frac{t}{t_{0}}\right)}^{-\left(b+1\right)}=Ce^{\frac{-U_{eff}}{k_{B}T}}=Ce^{\left[ln{\left(\frac{J^{*}}{J}\right)}^{\frac{-U_{0}}{k_{B}T}}\right]}=C{\left(\frac{J^{*}}{J}\right)}^{\frac{-U_{0}}{k_{B}T}}$$

Introducing the (experimental) time-dependent probing current from Equation (5) into Equation (9) leads to the equality
(10)$$-bJ_{0}t_{0}^{-1}{\left(\frac{t}{t_{0}}\right)}^{-\left(b+1\right)}=C{\left(\frac{J^{*}}{J_{0}}\right)}^{\frac{-U_{0}}{k_{B}T}}{\left(\frac{t}{t_{0}}\right)}^{\frac{-bU_{0}}{k_{B}T}}$$
which must be true for any value of the variable time, hence, the exponents of (*t*/*t*_0_) in both sides are equal: −(*b* + 1) = −*b*(*U*_0_/*k*_B_*T*), or
(11)$$U_{0}=k_{B}T\left(1+\frac{1}{b}\right)$$

Equation (11) shows clearly that the pinning potential can be easily extracted from the slopes of the linear dependence (in the double-logarithmic plots) of our experimental data as shown in Figure 2 and Figure 3. Assuming Boltzmann constant to be unity, the results will be given very practically in Kelvin.

We need to stress the fact that Equation (4) describes the experimental data (straight line in the double-logarithmic plots in Figure 2 and Figure 3), which, as we have proven in this section, is compatible with the logarithmic dependence of the pinning potential on the probing current that was proposed by Zeldov and collaborators after experiments (magneto-resistance, I–V curves) on YBCO films. We are not aware of any other superconducting material with similar behavior to be reported in the literature, but, if similar experiments lead to a straight line in the *ln**J*_c_ vs. *ln*(*f*_0_/*f*) dependence, then the Zeldov model might be the most appropriate.

### 3.4. Experimentally-Determined Pinning Potentials

All plots of experimental data similar to those shown in Figure 2 and Figure 3, for all samples and in all the DC magnetic fields in which we found a maximum in the χ″(*h*_ac_) dependences, showed the same linear dependence in the double logarithmic plots, and, by using Equation (11), we could estimate the pinning potentials. As previously discussed, for thinner films (0.3 μm and 0.4 μm), the range of applied DC fields for which a maximum in χ″(*h*_ac_) dependences was found is larger, while for thicker films, only a few values of DC fields allowed such estimations. In the following, we are presenting the experimental data of *J*_c_ (in logarithmic scale) as function of *ln*(*f*_0_/*f*), together with the fits of the data with Equation (4), for 6 of the measured samples. From the fits of the data, the slopes *b* in the dependence given by Equation (4) was used to estimate the pinning potential *U*_o_, using Equation (11). The six representative samples showed below are: a 0.3 μm thin film of YBCO doped with 4% BZO (Sample 1, Figure 4a); a 0.4 μm thin film of YBCO doped with 4% wt. BZO and 2% at. Y_2_O_3_ (Sample 2, Figure 4b); a 1.1 μm thick film of YBCO doped with 4% wt. BZO (Sample 3, Figure 5a); a 1.1 μm thick film of YBCO doped with 4% wt. BZO grown on substrate decorated with Ag nano-islands by using 15 laser shots on the silver target (Sample 4, Figure 5b); a quasi-bilayer of total thickness of 2.2 μm composed of two YBCO layers doped with 4% BZO separated by a quasi-layer of Ag nano-islands (15 laser shots on silver target), grown on Ag nano-islands decorated substrate (Sample 5, Figure 6a); and, finally, a bilayer of total thickness of 3 μm composed of two YBCO layers doped with 4% BZO separated by a nano-layer of STO (15 nm thickness), grown on Ag nano-islands decorated substrate (Sample 6, Figure 6b). The experimental data, together with the fits, are shown in Figure 4, Figure 5 and Figure 6 below.

It can be clearly seen that, in the case of thin films (0.3–0.4 μm, Samples 1 and 2) in Figure 4, there is a large range of fields in which we could find a maximum in the AC field dependence of the out-of-phase susceptibility, while for the thicker films (1.1 μm, Samples 3 and 4) in Figure 5, the range of DC fields in which the maximum was detected is much smaller.

From the data fitted with Equation (4), the slopes *b* were obtained, and, from Equation (11), with *T* = 77 K, pinning potential values *U*_o_ were obtained (in K, *k*_B_ = 1), and presented in Table 1.

Due to the limitations in the availability of higher magnetic fields, we were able to perform only a few similar measurements at 65 K and fields between 11 and 14 T. The experimental data (not shown) display the same linear dependence in the double-logarithmic plots. The resulted values of the pinning potentials at 65 K are shown in Table 2.

As a guide to the eye, the estimated values of the pinning potential *U*_0_ showed in Table 1 and Table 2 are also plotted in Figure 7, for both temperatures, 77 K (a) and 65 K (b).

## 4. Discussion

Although the proposed method is very straightforward and effective in estimating the frequency (time) dependent critical current density, for a given temperature of interest, the DC field range of applicability is limited by the maximum available amplitude of the AC excitation field, which, in commercially-available measurement systems (e.g., *Quantum Design* PPMS, ACMS option), is below 16 Oe. Obviously, the positions of the maxima in χ″(*h*_ac_) dependences (as in Figure 1) have to be at values lower than 16 Oe; thus, only critical current densities below a certain value (which is thickness-dependent) can be estimated. As can be seen in Figure 4a,b, for ***thin*** films (0.3 or 0.4 μm), at 77 K the windows of useful DC fields are quite large, 4–5 T, while for ***thick*** films (as in Figure 5 and Figure 6), the window of useful DC field is only 2, maximum 3 T, at fields larger than 5–6 T. If such large fields are not available, the reduction of the critical current density below the measurable threshold has to be performed by increasing the measurement temperature closer to the critical temperature.

The dependence of the critical current density estimated from the positions of the maxima in χ″(*h*_ac_) curves on the reduced frequency of the AC excitation field proved to be straight lines in a double-logarithmic plot, which we have shown to be compatible with a logarithmic dependence of the pinning potential on the probing current density, while attempted fits with the two “classical” models of flux pinning deviate significantly from the experimental data, as can be seen from Figure 2 and Figure 3.

The experimentally-determined slopes in the *J*_c_ (in logarithmic scale) vs. *ln*(*f*_o_/*f*) can be related (although not quite straightforward, but through a careful mathematical derivation) to the pinning potential, which was successfully estimated for all six samples, for the DC fields in the useful ranges, at 77 K and, in a few cases, at 65 K. The data shown in Table 1 and Table 2 revealed values of *U*_o_ much larger than the experimental temperatures (77 or 65 K), and, considering that the probability of flux lines jumping/creeping out of the potential well is inversely-proportional to the exponential of *U*_o_/*T*, this probability is very small. For example, if we look for the data for Sample 1, at 77 K and 6 T, *U*_o_/*T =* 524/77 ≈ 6.8, while exp (−6.8) ≈ 10^−3^, while at 65 K and 14 T, *U*_o_/*T =* 456/65 ≈ 7, and exp (−7) ≈ 9 × 10^−4^. Such very low probabilities of flux creep/jump outside the artificially nano-engineered pinning centers are obviously reflected in the very high values of the critical current densities, even in high applied DC fields, a very desirable property of such materials for power applications in high magnetic fields.

A closer look at the data in Figure 4, Figure 5 and Figure 6, and in Table 1 and Table 2, allows us to draw a few qualitative conclusions: (i) it is obvious that both critical current densities and pinning potentials are larger in the thin films (Samples 1 and 2), consistent with the well-known fact that critical current density in YBCO films as function of film thickness has a peak at around 0.3–0.4 μm. However, for practical applications the property of interest is not critical current density, but the critical current per cm-width of the tape. The increase of the film thickness without a strong decrease of its critical current density is a matter of practical interest, and this can be achieved by substrate decoration with Ag nano-islands which has a catalytic effect (Sample 4) and/or the use of quasi-multilayer or multi-layer approach (Samples 5 and 6). (ii) Apart from Samples 1 and 3 which contain artificial pinning centers induced by BZO, all the other samples in this study have synergetic pinning centers, which are very useful in increasing the critical current density not only for magnetic fields parallel to the *c*-axis, but also for other magnetic field orientation. Transmission Electron Microscopy (TEM) images of samples similar to those studied in this work were presented elsewhere: self-assembled nanoscale defects (pinning centers) induced by BZO and/or Ag nano-islands (Samples 1, 3–5) can be found in Refs. [12,13,18]; nanoscale defects induced by BZO and Y_2_O_3_ can be seen in Ref. [8]; and nano-scale defects induced by BZO, Ag nanoislands, and STO nanolayers can be seen in Ref. [14]. A comparison of the results in this work with pinning potentials of YBCO superconducting films with various artificial pinning centers reported elsewhere in the literature is quite difficult to make since, as we stressed in the introduction, the vast majority of the work in this field report improvement of critical current density determined mainly from transport measurements and/or magnetic hysteresis loops, and with the improvement on bulk pinning force extracted from the same magnetization loops, and not the average pinning potential of individual pinning sites. The technique presented in this paper has not been widely used up until now. Pinning potential can also be estimated from magnetic relaxation measurements, which have been reported for four of the samples presented in this work. Samples number 1 and 2 (thinner films) have a pinning potential, at 77 K, in 0.2 T, of 1500–1700 K [26], while samples 3 and 4 (thicker films) have a pinning potential of about 2000 K [27] in the same conditions of temperature and DC field (77 K, 0.2 T). The method described in this work is suitable to larger fields, but, looking at the date in Table 1, the results in this work and those from magnetic relaxation are not contradictory. As stressed in the introduction, most articles in the literature report bulk pinning force (in GN/m^3^) obtained from DC magnetic hysteresis measurements, which is an indication of the pinning strength for the whole sample, while the method presented in this work is an indication of the average pinning potential of individual pinning sites. The technique presented in this work can be a useful tool in designing self-assembled artificial pinning centers with higher potential. However, the optimum architecture of the pinning centers depends strongly on the conditions for operation (temperature, DC magnetic field) and the scope of the application, meaning that this kind of study should be coupled with other types of measurements (e.g., transport measurements with variation of angle between the film and the field), since our method presented in this paper gives indications only for in-plane currents in fields perpendicular to the films.

## Figures and Tables

**Figure 1 nanomaterials-12-01713-f001:**
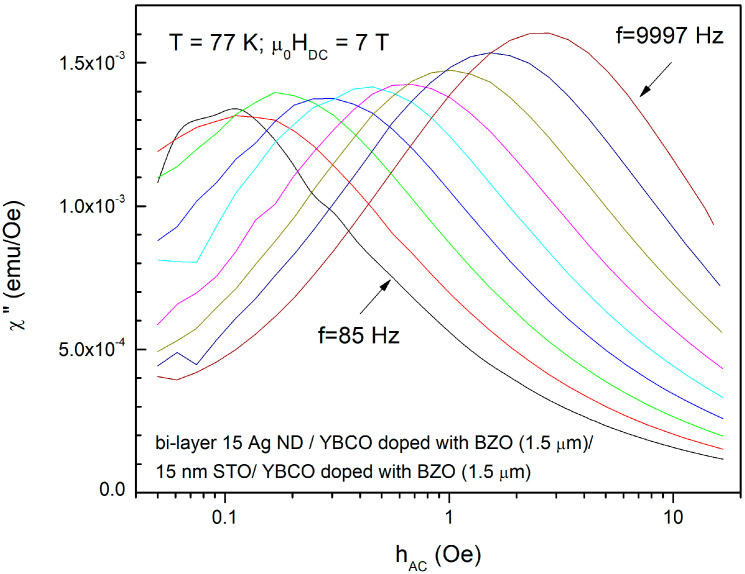
AC field amplitude *h*_ac_ dependence of the out-of-phase susceptibility χ″ of the bi-layer film composed of two BZO-doped YBCO films of 1.5 μm thickness, grown on substrates decorated with Ag nano-islands, separated by a nanolayer (15 nm) of insulating STO, at 77 K, in a DC field of 7 T, and at various frequencies (different colors) between 85 and 9997 Hz.

**Figure 2 nanomaterials-12-01713-f002:**
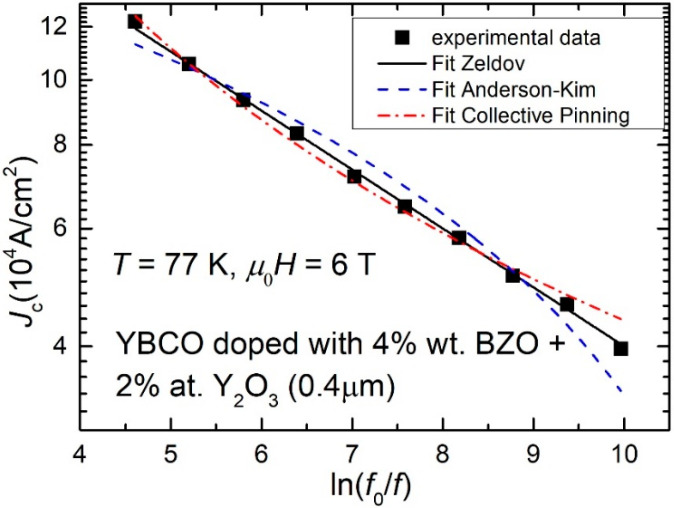
Critical current density *J*_c_, at 77 K and in a DC field of 6 T, of a 0.4 μm-thick film of YBCO doped with 4% wt. BZO and 2% at. Y_2_O_3_, obtained from measurements similar to those exhibited in Figure 1 by using Equation (1), as function of AC field frequency *f*, in a double-logarithmic scale. *f*_0_ in the reformulation of the x-axis is a macroscopic attempt frequency of flux jumps outside the pinning centers of about 10^6^ Hz [25]. The dash line (blue) is the best fit with the Anderson–Kim model (Equation (2)), the dash-dot line (red) is the best fit with the collective pinning model (Equation (3)), while the best fit in this double logarithmic plot is a straight line (full line—black), which is compatible with the logarithmic dependence of the pinning potential on probing current proposed by Zeldov et al., as we will prove in Section 3.3.

**Figure 3 nanomaterials-12-01713-f003:**
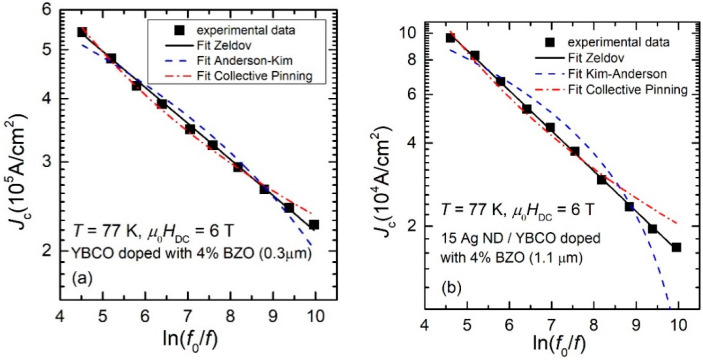
Dependence of the critical current density *J_c_* on the frequency of the AC excitation field, at 77 K and in a DC field of 6 T, for a 0.3 μm-thick film of YBCO doped with 4% wt. BZO (**a**) and for a 1.1 μm-thick film of YBCO doped with 4% wt. BZO grown on a STO substrate decorated with Ag nanodots obtained by 15 laser shots on the silver target (**b**). All other details are similar to Figure 2.

**Figure 4 nanomaterials-12-01713-f004:**
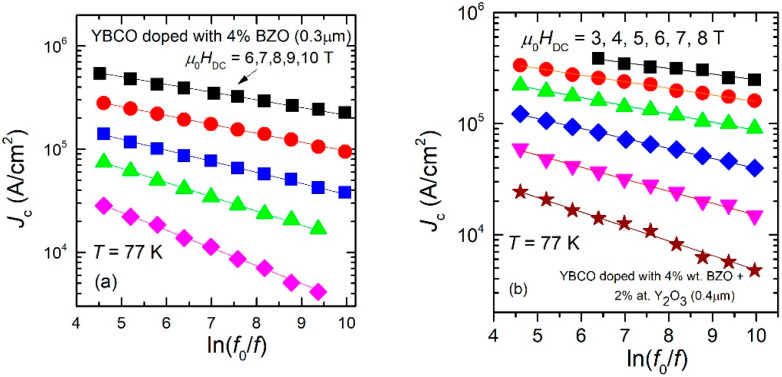
Frequency dependence of the critical current density of Sample 1 (**a**), and of Sample 2 (**b**) at 77 K and in DC fields indicated in the figure. All the dependences are very well described by straight lines in the double-logarithmic plot.

**Figure 5 nanomaterials-12-01713-f005:**
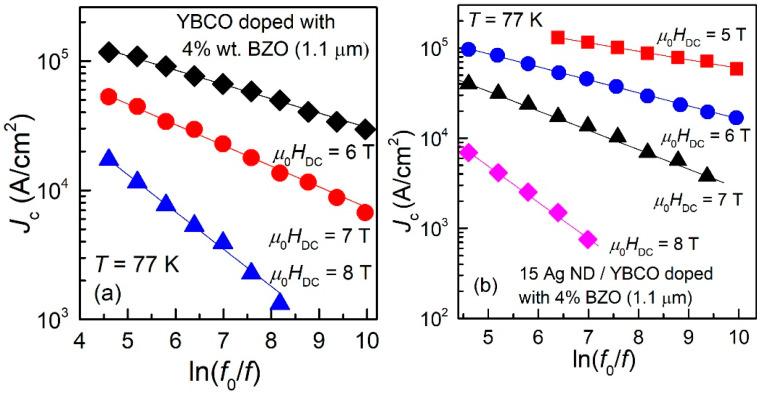
Frequency dependence of the critical current density of Sample 3 (**a**), and of Sample 4 (**b**) at 77 K and in DC fields indicated in the figure. All the dependences are very well described by straight lines in the double-logarithmic plot.

**Figure 6 nanomaterials-12-01713-f006:**
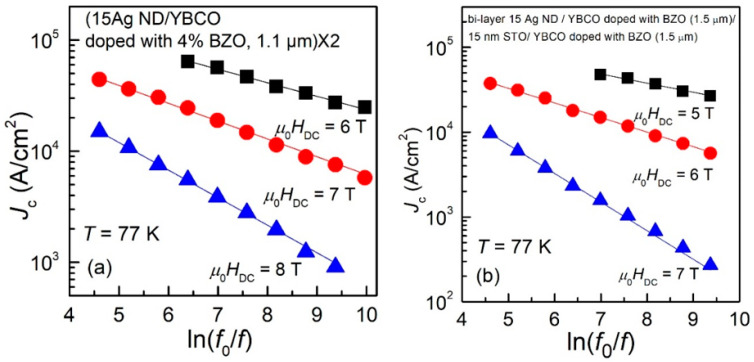
Frequency dependence of the critical current density of Sample 5 (**a**) and Sample 6 (**b**), at 77 K and in DC fields indicated in the figure. All the dependences are very well described by straight lines in the double-logarithmic plot.

**Figure 7 nanomaterials-12-01713-f007:**
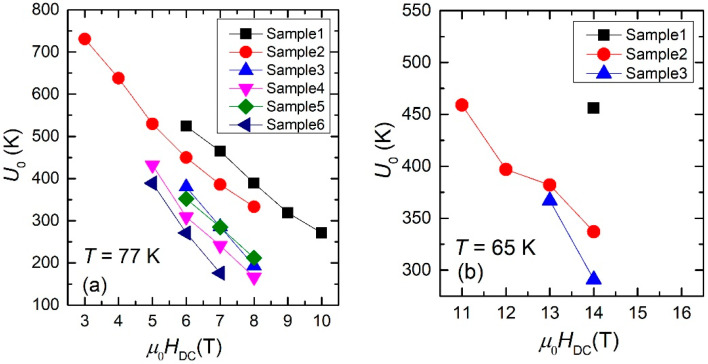
Estimated values of the pinning potential in various DC fields. (**a**) T = 77 K, (**b**) T = 65 K.

**Table 1 nanomaterials-12-01713-t001:** Values of pinning potential estimated from slopes *b* in Figure 4, Figure 5 and Figure 6, for Samples 1 to 6, in various DC magnetic fields, at 77 K.

Sample	DC Field (T)	*U*_o_ (K) at *T* = 77 K
Sample 1	6	524
	7	465
	8	389
	9	319
	10	271
Sample 2	3	731
	4	638
	5	530
	6	450
	7	386
	8	333
Sample 3	6	380
	7	286
	8	193
Sample 4	5	432
	6	309
	7	241
	8	166
Sample 5	6	352
	7	285
	8	212
Sample 6	5	389
	6	271
	7	176

**Table 2 nanomaterials-12-01713-t002:** Values of pinning potential estimated from slopes *b*, in various DC higher magnetic fields, at 65 K.

Sample	DC Field (T)	*U*_o_ (K) at *T* = 65 K
Sample 1	14	456
Sample 2	11	459
	12	397
	13	382
	14	337
Sample 3	13	367
	14	291

## Data Availability

The relevant data for the discussion are completely supplied in the Results section. Raw data are available upon request.

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
