# Peer review of "Pinning Potential of the Self-Assembled Artificial Pinning Centers in Nanostructured YBa2Cu3O7−x Superconducting Films"

_nanomaterials, 2022, doi:10.3390/nano12101713_

Round 1

Reviewer 1 Report

The authors try to propose a convenient approach of estimating the pinning potential of the Y-123 based films by the AC susceptibility measurements. The data in the manuscript provide a convincing feasibility of the method. However, some questions need to be answered before the publication:

1. Page 6 Line 234, the approach of estimating the pinning potential is based on a linear dependence of JC in logarithmic scale on ln(f0/f). It is very important to know in what conditions can the equation 4 be applied. Is it only available in Y-123 based films? Please provide some further statement about range of the applicability of equation (4).

2. Page 10 Line 338 to Page 11 Line 339, according to Figure 2 and 3, both fittings with Anderson-Kim model and collective pinning model do not match the experimental data. The fitting with Anderson-Kim model present further deviation from the data. But is it really convincing to say the data are better described by the collective pinning model? Can the authors provide some proof to the statement?

3. Since the authors have made a comparison of some samples, is it possible to provide some suggestion about how to design the artificial pinning centers to achieve higher pinning potential?

Reviewer 2 Report

Adrian Crisan et al. reported the dependence of the out-of-phase susceptibility on the amplitude of the AC excitation magnetic field. From these measurements, the critical current density versus frequency was plotted and the activation energy was reported. Several models were used to explain the critical current density dependences. Overall, the text is well-written. However, there are important points to be clarified and some improvements are needed:

1. The title is not appropriate. The authors should change the title by specifying the type of artificial pinning centers reported in this work.

2. The authors should add the sources of the used materials.

3. The authors should add schematic illustration for their sample’s preparation

4. In the experimental section: the authors should give the values of fixed temperature and fixed Dc field used for the measurements.

5. The authors should specify the geometry of their samples used for the determination of JC.

6. Why do the authors represent Fig.1 in a double logarithmic scale?

7. Line 201: The authors stated “Measurements similar to those in Figure 1 were performed for all the analyzed samples, in various applied DC fields” Were the measurements done in various applied DC fields or at various applied AC fields? Please clarify this statement.

8. A discussion regarding the comparison of pinning potentials between the 6 samples we have analyzed in this paper is also very useful. This statement should be revised and typo mistakes should be corrected.

9. There is no comparison between the results obtained and the results of recent work in this work. Authors should compare their results with the literature.

10. TEM or SEM images should be provided to confirm the role of pinning centers in the flux pinning properties of superconductors. 
